# Strategies to Re-Sensitize Castration-Resistant Prostate Cancer to Antiandrogen Therapy

**DOI:** 10.3390/biomedicines11041105

**Published:** 2023-04-06

**Authors:** Belén Congregado Ruiz, Inés Rivero Belenchón, Guillermo Lendínez Cano, Rafael Antonio Medina López

**Affiliations:** Urology and Nephrology Department, Biomedical Institute of Seville (IBIS), University Hospital Virgen del Rocío, 41013 Seville, Spain

**Keywords:** castration-resistant prostate cancer, antiandrogen therapy, resensitizing to antiandrogens, bipolar antiandrogen therapy, androgen receptor

## Abstract

Since prostate cancer (PCa) was described as androgen-dependent, the androgen receptor (AR) has become the mainstay of its systemic treatment: androgen deprivation therapy (ADT). Although, through recent years, more potent drugs have been incorporated, this chronic AR signaling inhibition inevitably led the tumor to an incurable phase of castration resistance. However, in the castration-resistant status, PCa cells remain highly dependent on the AR signaling axis, and proof of it is that many men with castration-resistant prostate cancer (CRPC) still respond to newer-generation AR signaling inhibitors (ARSis). Nevertheless, this response is limited in time, and soon, the tumor develops adaptive mechanisms that make it again nonresponsive to these treatments. For this reason, researchers are focused on searching for new alternatives to control these nonresponsive tumors, such as: (1) drugs with a different mechanism of action, (2) combination therapies to boost synergies, and (3) agents or strategies to resensitize tumors to previously addressed targets. Taking advantage of the wide variety of mechanisms that promote persistent or reactivated AR signaling in CRPC, many drugs explore this last interesting behavior. In this article, we will review those strategies and drugs that are able to resensitize cancer cells to previously used treatments through the use of “hinge” treatments with the objective of obtaining an oncological benefit. Some examples are: bipolar androgen therapy (BAT) and drugs such as indomethacin, niclosamide, lapatinib, panobinostat, clomipramine, metformin, and antisense oligonucleotides. All of them have shown, in addition to an inhibitory effect on PCa, the rewarding ability to overcome acquired resistance to antiandrogenic agents in CRPC, resensitizing the tumor cells to previously used ARSis.

## 1. Introduction

Prostate cancer (PCa) is the most common tumor diagnosed in men in the Western World, the third leading cause of cancer death in Europe, and the second in the United States [1,2]. Most PCa is diagnosed at the localized stage where active surveillance, surgery, and radiotherapy achieve very high disease control rates; however, relapsing, locally advanced, and metastatic cancer require systemic treatment. In this setting, targeting the androgen receptor (AR) signaling axis, whose activation is mediated by androgens, is the key to controlling prostate cancer progression [3]. In that sense, androgen deprivation therapy (ADT) with luteinizing hormone-releasing hormone (LHRH) agonists, LHRH antagonists, antiandrogens, or surgical castration is initially recommended [4] and typically leads to a 90–95% decrease in circulating androgen levels and prevents tumor cell survival. However, invariably over time, cells develop resistance to this state of androgen deprivation, and the tumor becomes castration-resistant, which is defined as a PCa that has progressed despite castrate levels of serum testosterone (<50 ng/dL) [5].

Castration-resistant prostate cancer (CRPC) had been considered “androgen-independent” or “hormone-refractory” in the past, but today, we know that acquiring castration-resistant status does not mean tumor cells are nonresponsive to hormone inhibition. In fact, CRPC remains highly dependent on the AR signaling axis, so it is recommended to maintain ADT indefinitely as the mainstay of treatment [6,7]. Furthermore, this AR signaling reactivation is demonstrated by the effectiveness of the new potent antiandrogens used in this setting.

The exact mechanisms underlying the transition from androgen-dependent PCa to CRPC are yet to be fully understood. AR reactivation in CRPC occurs despite the androgen deprivation induced by ADT, and although the main mechanism triggering CRPC is through the AR signaling pathway, AR-independent pathways have also been detected [8,9]. Several mechanisms causing progression to CRPC have been described: (1) AR gene amplification and overexpression; (2) somatic AR mutations; (3) prostate intracrine androgen biosynthesis; (4) AR splice variants (AR-Vs); (5) non-canonical AR transactivation; (6) AR-independent bypass pathways; (7) AR-negative cell populations: PCa stem-like cells; and (8) AR-negative cell populations: neuroendocrine PCa cells and other subtypes [10].

Since the beginning of this century, many drugs with different mechanisms of action have been approved and managed to prolong advanced PCa patient survival. Clinical options include: (1) cytostatic agents (docetaxel plus prednisone, cabazitaxel); (2) newer-generation androgen receptor signaling inhibitors (ARSis) such as the AR antagonists enzalutamide, darolutamide, and apalutamide, or the androgen biosynthesis inhibitor abiraterone acetate; (3) bone-targeted therapies (Radium 223); (4) immunotherapy (sipuleucel-T); (5) radioisotopes (lutetium-177–PSMA-617) and (6) poly ADP-ribose polymerase (PARP) inhibitors (olaparib and rucaparib) [11]. Incorporating these potent drugs at increasingly earlier stages of the disease results in much better overall survival rates [4]. However, this intense pressure on cancer cells leads them to search for adaptive survival mechanisms. In fact, stronger inhibition of AR signaling produced by potent second-generation antiandrogens induces an increasing number of AR-negative metastatic PCa, no longer responsive to antiandrogens. This type of tumor is rare in untreated patients; however, due to the wider use of ARSis in earlier stages of the disease, its incidence is increasing [12,13].

Therapeutic resistance not only occurs in prostate cancer, but it is a constant challenge in oncology. Castration resistance results from adaptive pathways created by PCa cells to elude current therapies. By identifying these pathways and inhibiting them, researchers could overcome treatment resistance and significantly improve patient survival and quality of life. Research on advanced PCa specifically focuses on the following:-Finding new drugs with different mechanisms of action from those previously used;-Using combination therapies to boost synergies, enhancing the action of each drug;-Research treatments that can resensitize tumor cells to previously used mechanisms of action.

In this sense, it may be possible to resensitize tumor cells to antiandrogens, maintaining or restoring the dependence of the AR signaling axis. This re-expression of AR in AR-negative cells could revitalize the activity of AR inhibitors.

This rechallenge is a well-known strategy in oncology. It consists of using again an oncological line that was considered exhausted on initially suitable responder patients. In this article, we will review all those strategies and drugs that have been demonstrated to be able to resensitize CRPC cells to previously used antiandrogens. The one with the strongest scientific evidence is bipolar androgen therapy (BAT).

BAT is a concept that was born years ago but is now being established as an emerging treatment strategy for patients with metastatic CRPC (mCRPC). This method involves cycling serum testosterone from supraphysiologic down to near-castrate levels every month [14]. Although many clinical trials have demonstrated the benefit of BAT as a single-agent strategy in mCRPC, in this article, we will mainly focus on the interesting finding of its ability to resensitize patients to prior novel AR-targeted therapies.

A recently published systematic review focuses on the efficacy and safety of BAT in mCRPC after abiraterone or enzalutamide [15]. However, our review will emphasize the ability of BAT to restore sensitivity to abiraterone and enzalutamide and also will show other drugs that have proven a similar ability.

These other compounds have demonstrated their activity against CRPC cells in vitro and in vivo. Though, in some cases, this effect has yet to be shown in patients on monotherapy, and in other cases, the agent results in unacceptable toxicity. However, all these molecules show a promising alternative pathway that would involve sensitizing to prior antiandrogen treatments or reversing intrinsic or acquired resistance to second-generation antiandrogens [16]. This is why we will also review all these agents.

This article aims to review the mechanisms that allow cancer cells to switch from an AR-positive to an AR-negative disease, focusing on potential strategies to overcome these mechanisms and successfully resensitize these CRPC cells to antiandrogen therapy. BAT has proved to be the most effective of all antiandrogen resensitization mechanisms. Therefore, we will review the most recent literature on BAT and will provide interesting data on other agents, such as niclosamide, indomethacine, metformine, lapatinib, panobinostat, and ipatasertib.

To the best of our knowledge, this is the first article to extensively review this interesting antiandrogen resensitization or rechallenge strategy.

## 2. Evidence Acquisition

In January 2023, a comprehensive database search was performed based on PICO (Population, Intervention, Comparator, Results) criteria [17,18]. Studies published up to this date were included and a narrative review was conducted (Figure 1).

The population consisted of castration-resistant prostate cancer cells (P). Treatment of interest was androgen deprivation therapy (I). We considered eligible single or multiple arms studies, so no comparator was mandatory (C). Main outcomes of interest were the mechanism of castration resistance and strategies to resensitize cells to antiandrogen therapies (O).


Studies identification: 42 articles using the terms “resensitizing to antiandrogen” AND “prostate cancer” were found. Out of them, 6 were reviews and systematic reviews, while 36 were original articles. The consistency of this revision is affected by the inherent lack of robust evidence in urology.Screening: No articles were duplicated, so 42 articles were screened by title and abstract. Out of them, 24 full-text articles were assessed for eligibility.Eligibility: The selection criteria were (a) reviews, original articles, and preclinical studies, (b) studies about castration resistance mechanisms and strategies to resensitize cells to androgen deprivation therapy. Exclusion criteria were (a) non-English/Spanish literature, (b) editorials, comments, and letters, and (c) studies published before 2012, as they were not updated enough for our purpose.Study analysis: After applying the above eligibility criteria, 19 studies about castration resistance mechanism and strategies to resensitize cells to androgen deprivation therapy were selected. In total, 16 articles with the highest level of relevance to the discussed topics were selected with the consensus of the authors. A review from the selected studies was conducted.


## 3. The Androgen Receptor

AR is a protein encoded by the AR gene, which is located on the X chromosome at Xq11-12. It has several functional domains: (1) NTD (terminal transactivation domain, necessary for cellular transcription); (2) DBD (DNA-binding domain, responsible for the binding of the AR protein to specific DNA sequences); (3) the hinge region (encoding the nuclear translocation signal); and (4) LBD (ligand-binding domain, which binds androgen ligands) [10,19].

AR is a ligand-dependent transcription factor whose main function is to respond to androgenic steroid hormones, such as testosterone and DHT. After binding them, it causes a conformational change in AR that originates a nuclear translocation and induction of androgen-responsive gene expression, which allows the development of prostate cells [20].

Like normal prostate cells, PCa cells require androgens for continued growth. ADT reduces testosterone levels by 90–95% and is initially effective in PCa patients by suppressing gonadal testosterone production. Nevertheless, intraprostatic levels of DHT decline by only about 50%.

Stimulation of intraprostatic DHT and other androgens, such as those from the adrenal gland, can be blocked by adding a first- or second-generation antiandrogen drug.

In the absence of one of these androgenic ligands, testosterone or DHT, AR is sequestered in the cytoplasm where it is inactive, but in a conformation with a high ligand-binding affinity.

Inhibition of the AR induces both cell cycle arrest and apoptosis of PCa cells [8].

Following androgen binding to the LBD of the AR, the receptor leaves the cytoplasm and translocates to the nucleus, where it dimerizes with a second AR to regulate the transcription of androgen-dependent target genes. Transcriptional regulation of these target genes through persistent RA signaling contributes to PCa proliferation and survival.

## 4. Castration Resistance Mechanisms

CRPC is defined as cancer that progresses despite effective suppression of testosterone below castrate levels (≤50 ng/dL) and encompasses a broad spectrum, from an asymptomatic localized disease where the only sign of progression is an elevated prostate-specific antigen (PSA) to rapidly progressive metastatic disease that leads to death.

The term “castration resistant” is justified by the intracrine production of androgens, which is at least partially responsible for ADT resistance [21].

The reduction in circulating testosterone levels caused by ADT causes the AR to remain inactive and, thus, no longer activates androgen-dependent target genes that drive the PCa viability and proliferation.

However, AR reactivation, despite effective suppression of testosterone below castrate levels, leads to a CRPC phenotype in all patients with PCa treated with ADT, with or without antiandrogens. Other mechanisms independent of AR also drive CRPC [22,23].

### 4.1. AR-Dependent Mechanisms Triggering CRPC

#### 4.1.1. AR gene Amplification and Overexpression

One mechanism of AR reactivation in CRPC is increasing AR expression through genomic amplification of the AR locus or upregulation of AR protein levels. In both cases, increased AR expression can sensitize PCa cells to low levels of androgens [24].

This overexpression of the AR mRNA is the most frequent alteration in CRPC (up to 81% of cases) [25,26].

#### 4.1.2. Somatic AR Mutations

Studies have shown a range of AR mutations in androgen-dependent tumors from 2% to 25%, while the incidence in CRPC tumors is around 10–40% [27].

These AR mutations may confer hypersensitivity to androgens, broaden ligand specificity [28], or lead to a promiscuous activation of AR in response to atypical ligands such as adrenal androgens, other steroid hormones, or antiandrogen drugs.

The most frequent AR mutation is T878A, which correlates with decreased specificity of the AR ligand, allowing the receptor to be activated by other molecules, such as estrogens and glucocorticoids [29].

This mutation occurs more frequently in response to drugs targeting androgen synthesis, such as abiraterone [30], and this fact may be relevant because patients on abiraterone treatment need to be treated simultaneously with prednisone to counteract some adverse effects. Similarly, some androgen precursors, such as pregnenolone and progesterone, accumulate during abiraterone treatment, and some of these have also been identified to bind mutated AR and instigate downstream AR signaling [31].

#### 4.1.3. Prostate Intracrine Androgen Biosynthesis

When there is testicular testosterone deficiency due to ADT, PCa cells themselves are capable of transforming adrenal cholesterol and androgen precursors into testosterone to activate AR signaling.

These cells have been shown to contain all the essential components to carry out androgen biosynthesis, and an elevation of many of the key enzymes to carry out this synthesis in recurrent or metastatic tumors has been objectified [32].

#### 4.1.4. AR Splice Variants (AR-Vs)

AR splice variants (AR-Vs) are caused by alternative gene splicing or rearrangement of the AR gene, which results in AR LBD loss [33,34].

Approximately 20–40% of patients present with primary resistance to second-generation antiandrogens, such as enzalutamide and abiraterone [35], and even worse, patients with an initial PSA response will develop secondary resistance to the drug [36,37].

Expression of these AR variants is strongly associated with resistance to both abiraterone and enzalutamide, the most widely studied of these variants, AR-V7, being particularly important, and is associated with worse results in these patients [38,39].

#### 4.1.5. Non-Canonical AR Transactivation

In the absence of androgens or when androgens are present at low levels, numerous growth factors, cytokines, and hormones may activate AR [40]. Some of these are insulin-like growth factor-1 (IGF-1) and interleukin-6 (IL-6) [41].

For example, cyclin D1b, an isoform that does not possess an AR inhibitory function, is elevated in clinical PCa specimens [42].

Similarly, the retinoblastoma (Rb) tumor suppressor gene induces an increase in AR mRNA and protein levels, which promotes castrate-resistant progression in PCa cells [43].

The aberrant activation of PI3K-Akt may induce PCa proliferation since the PI3K-Akt-mTOR pathways regulate important cellular processes, such as cell growth and programmed cell death [44]. Loss of PTEN tumor suppressor gene, a negative regulator of the PI3K/Akt pathway [45], has been shown to be associated with the development of antiandrogen resistance [46,47].

When AR interacts with PI3K subunits, the Akt pathway is activated, and therefore the cell proliferation [48].

In fact, it has been demonstrated that synergistic targeting of the PI3K/Akt pathway and the androgen receptor axis delays CRPC progression [49,50].

### 4.2. AR-Independent Mechanisms Triggering CRPC

#### 4.2.1. AR-Independent Bypass Pathways

It has been described that alternative signaling pathways allow the proliferation of CRPC bypassing AR. These AR bypass pathways remain active even in the absence of AR expression [51].

When AR signaling is blocked by antiandrogens, those clones that can bypass the AR are selected. An example is the glucocorticoid receptor (GR), which has a DNA-binding domain similar to that of the AR.

The GR may be able to bind to androgen response elements (AREs) and drive PCa cell survival under certain circumstances [52]; for example, in patients treated with chemotherapy together with second-generation antiandrogens.

#### 4.2.2. AR-Negative Cell Populations: PCa Stem-Like Cells

According to the cancer stem cell or tumor-initiating cells (TICs) theory, only a population of tumor cells that possess stem cell properties can trigger tumor initiation [53,54].

Cancer stem cells are resistant to most therapies and are capable of developing a tumor with increased metastatic capacity [55].

Several populations of PCa stem-like cells (PCSCs) have been described, such as CD44+/α2β1+/CD133+, TRA-1-60/CD151/CD166, ALDH, and PSA-/lo, for example. It appears that PCSCs are AR-negative or express very low levels of AR, and, therefore, tumors consisting of these cells would have a poor response to antiandrogens and other AR-targeted therapies [56,57].

The mechanism that would explain the relationship between PCSC and CRPC would involve the expansion of a residual AR-negative/low stem-like cell population remaining after androgen deprivation and potentially repopulating tumors with castrate-resistant tumor cells [58].

#### 4.2.3. AR-Negative Cell Populations: Neuroendocrine PCa Cells and Other Subtypes

Neuroendocrine prostate cancer (NEPC) is the most common type of AR-negative tumor [59]. Neuroendocrine (NE) cells are scarcely represented in a normal prostate gland and are characterized by a loss of expression of AR-regulated markers [60]. Histologically, it comprises a heterogeneous group of tumors [61].

NEPC can be classified as de novo or treatment-related. De novo NEPC is rare, typically less than 2% of all primary PCa, and very aggressive [62]. More frequently, NEPC comes from an adenocarcinoma previously treated with androgen deprivation therapy.

Interestingly, it has been shown that the neuroendocrine subtype is not entirely foreign to AR-positive cells and partly originates from AR-positive conventional adenocarcinoma through a transdifferentiation process.

Zou et al. have demonstrated that nearly all tumors with neuroendocrine markers have a luminal-epithelial origin [63]. Moreover, it has been shown that, when subjected to an androgen-depleted cell medium, androgen-sensitive prostate adenocarcinoma cells exhibit neuroendocrine differentiation, implying that castration also promotes NEPC development [64].

Bluemn et al., detected a subtype of PCa associated with ARSis treatment, whose incidence has increased from 5% to 20% in recent years, and that are negative for AR and neuroendocrine markers [12]. In addition, Labrecque et al. have described five different types of mCRPC based on RNA expression of AR and the most common NE markers [65].

Since the AR pathway is an essential therapeutic key in PCa, many researchers have focused on finding ways to restore AR signaling in AR-negative PCa. It would be interesting to find out the functional consequences of AR re-expression and if AR reactivation will suppress the oncogenic potential of AR-negative PCa cells revitalizing the activity of AR inhibitors.

Figure 2 summarizes mechanisms of resistance to androgen deprivation therapy.

## 5. Agents and Strategies to Resensitize CRPC to Antiandrogen Therapy

### 5.1. Bipolar Androgen Therapy

Of all the compounds and strategies currently under study to resensitize CRPC cells to antiandrogens, BAT accumulates the most scientific evidence and promising results.

BAT is based on the observation that the growth of some AR-expressing “androgen sensitive” human PCa cells can be inhibited by supraphysiologic levels of testosterone followed by a rapid drop of testosterone to castrate levels.

This “bipolar androgen therapy” will not allow time for PCa cells to adapt their AR expression in response to environmental conditions [14].

Different mechanisms have been proposed to explain this, including the induction of genomic instability through DNA double-strand breaks (DSBs) and the stabilization of the link between DNA and the AR, which prevents AR degradation and DNA relicensing during the cell cycle. Other postulated mechanisms are the modulation of oncogene expression, the inhibition of AR-V7 expression, and the reversal of transition to the neuroendocrine phenotype Figure 3.

Among the proposed mechanisms, one of the most important is that high-dose testosterone can induce double-strand DNA damage [66].

Supraphysiological concentrations of DHT induce DNA breaks and result in the activation of DNA damage response pathways, and these androgen-induced DSBs are independent of androgen-induced cell cycle progression [67].

When PCa cells are acutely exposed to supraphysiologic androgen levels, AR is saturated. This induces a paradoxical cancer cell death via over-stabilizing AR, preventing full relicensing in the next cell cycle [68].

Different proposed mechanisms of action are: the repression of SKP2 and MYC [69] and the suppression of circular RNA-BCL2 (circRNA-BCL2) expression by a very high dose of testosterone, which induces the autophagic cell death and avoiding CRPC cell growth [70].

Probably, there may be no single mechanism that explains the effectiveness of BAT, and several mechanisms of action coexist.

BAT involves giving sequential cycles alternating periods of acute supraphysiologic androgen followed by acute ablation.

The BAT concept uses the adaptive increase in AR that occurs in the CRPC cell after ADT to make it vulnerable to high androgen levels exposure (5 to 10 times higher than the normal testosterone level and about 100 to 200 times higher than the castration level), and after that lowers back the testosterone to castration levels.

In this way, it is possible to damage two types of CRPC cells: those with very high levels of AR, which will be sensitive to supraphysiological levels of testosterone, and those with very low levels of AR, which will be sensitive to very low levels of testosterone. See Figure 4.

The first studies on the concept of BAT emerged in the 1990s when it was observed that supraphysiological androgen levels could paradoxically suppress the growth of some PCa cell lines and xenograft models [71,72,73].

After that, some studies showed that it was an effective and safe treatment in patients with metastatic PCa and CRPC [74,75].

Subsequently, it was described that BAT could also resensitize to antiandrogen therapy [76].

As pioneers of this treatment, Denmeade et al. conducted a pilot study on 16 asymptomatic CRPC patients. They demonstrated the safety and efficacy of BAT in combination with etoposide, achieving objective responses defined as PSA decreases ≥50% vs. baseline and Response Evaluation Criteria in Solid Tumors (RECIST).

Two very interesting ideas can also be extracted from this study. The first one comes from the post hoc exploratory analysis of the effect of BAT on subsequent hormonal therapies. This analysis revealed that 12 of 13 patients who had received at least one antiandrogen prior to BAT treatment experienced a PSA response when retreated with AR-directed therapy after BAT. The other idea is that BAT can improve the quality of life of patients undergoing treatment [77].

The following paper published by this group included a larger number of patients and focused on the ability of BAT to resensitize to antiandrogen therapy.

RESTORE (REsensitizing With Supraphysiologic Testosterone to Overcome Resistance) was a phase II study designed to evaluate the activity of BAT in asymptomatic mCRPC patients progressing on AR-targeted therapy.

The most exciting conclusions of this study came from its multicohort design. It evaluated the responses to abiraterone or enzalutamide upon rechallenge after BAT.

The trial enrolled four cohorts of patients: Cohort A: 30 men with mCRPC who had progressed on enzalutamide, Cohort B: 30 men who progressed on abiraterone acetate, Cohort C: 30 on first-line castration-only therapy, and Cohort D: 20 mCRPC men with inactivating somatic or germline gene mutations.

The coprimary endpoints were number of participants with PSA drop ≥50% from preBAT level, and after enzalutamide or abiraterone acetate post-BAT from baseline [78].

The authors did not find statistically significant difference in PSA50 response to BAT between both cohorts, although one of the most interesting results of the study was that PSA50 responses to ARSi rechallenges were higher in the post-enzalutamide cohort (68% vs. 16%, *p* = 0.001).

The other interesting finding was that the time to progression following rechallenge was more than four months longer in the post-enzalutamide vs. post-abiraterone cohort (12.8 vs. 8.1 mo, *p* = 0.04). Moreover, AR-V7-positive patients got worse outcomes with this strategy.

This study demonstrated that BAT is more effective at resensitizing to enzalutamide than abiraterone [79].

The results also suggest that the strongest benefit of BAT is in restoration of sensitivity to AR antagonists rather than as a primary therapy [80].

The main goal of the TRANSFORMER (Testosterone Revival Abolishes Negative Symptoms, Fosters Objective Response and Modulates Enzalutamide Resistance) trial was to compare BAT vs. enzalutamide in asymptomatic CRPC men after progression on abiraterone. The impact of sequential exposure to AR agonists or antagonists was also assessed since crossover to the opposite treatment upon progression was allowed.

Regarding primary endpoint, clinical or radiographic progression-free survival (PFS), BAT was not superior to enzalutamide.

Nevertheless, the interesting finding was that the most significant benefit from BAT was seen in patients experiencing progression on prior abiraterone.

Approximately 40% of patients crossed over to the opposite treatment at progression.

Those who crossed to enzalutamide post-BAT experienced a significantly greater response compared to those who did not use BAT as a bridging therapy. Median time to PSA progression was 10.9 months in the first group vs. 3.8, PSA50 response was 78% vs. 25%, and objective response (OR) improved to 29% in the post-BAT cohort vs. 4%.

Although the trial failed to demonstrate superior PFS with BAT over enzalutamide in post-abiraterone CRPC, the most important finding from our point of view is that post-abiraterone, BAT can significantly improve the magnitude and duration of response to enzalutamide [81]; see Figure 5.

Other authors have also proven through multicenter studies with real-world evidence that BAT may play a role in CRPC resensitization after multiple treatment lines [82].

Although the core of this revision is to further examine the ability of BAT to resensitize mCPRC to antiandrogen therapy aiming to improve the efficacy and duration of response to BAT, ongoing studies are evaluating the combined treatments of BAT with other treatments, such as the DNA-repair inhibitor olaparib (NCT03516812), the bone-targeted radiation therapy radium 223 (NCT04704505), and carboplatin (NCT03522064).

Interestingly, BAT has also been shown to have a role in the sensitization of PCa cells to ionizing radiation (IR). It has been demonstrated that high doses of androgens have a synergetic effect when combined with ADT and radiotherapy, causing DNA damage and inhibiting PCa cells growth [67,83].

Other lines of research dive into the hypothesis that treatments that induce DNA double-strand breaks, such as BAT, may modify the immune microenvironment, improving subsequent immunotherapy outcomes [84].

AR has been shown to be expressed in some types of immune cells [85], and ADT has been shown to promote the suppression of the immune response [86].

Likewise, it has been shown that natural killer (NK) cells target androgen-dependent PCa stem-like cells [87].

All these findings strengthen the hypothesis that supraphysiological doses of androgens can affect the immune response. In fact, in vitro and in vivo studies in animals suggest that the combination of BAT with PD-1/PD-L1 checkpoint inhibitors can inhibit the growth of CRPC cells [88].

The COMBAT study was carried out to demonstrate this hypothesis. This is a phase 2 clinical trial designed to evaluate the clinical activity of BAT in mCRPC patients progressing on ARSis, followed by a combination treatment of BAT and the anti-PD1 nivolumab.

The combination met the pre-specified primary endpoint of confirmed PSA50 response, which was 40.0%, and the objective response rate (ORR) for patients with measurable disease was 23.8%.

BAT plus nivolumab was well tolerated, considering it was a heavily treated population, and durable responses were observed in a subset of patients [89].

Unfortunately, not all patients respond to BAT, and the degree and durability of the response are unpredictable. However, some men are exceptional responders. For this reason, like in other treatments, it is essential to identify prognostic and predictive biomarkers [90].

Moreover, it has been suggested that resistance to this treatment is mediated by changes in AR levels and activity [91].

Although there most likely are multiple molecular mechanisms responsible for the clinical response to BAT, one the researchers have studied recently is that BAT suppresses the oncogene MYC in responding patients [92].

Sena et al. showed that growth inhibition of prostate cancer models during BAT requires high AR activity, which is partly driven by the downregulation of MYC, a potent driver of growth and proliferation highly expressed in PCa [93].

The authors demonstrated that high pretreatment AR activity predicts a better clinical response and prolongs progression-free and overall survival for patients on BAT through the downregulation of MYC expression [94].

In addition, they used samples from combat clinical trial [89] patients looking for a response predictor by studying differences in AR activity before treatment through responders vs. non-responders. With this objective, they created an AR activity score using the Mann–Whitney ranking of expression of 10 canonical AR target genes, which they called the ARA_MW_ score.

Notably, responders had significantly higher preBAT ARA_MW_ scores than non-responders (*p* = 0.011), leading to the conclusion that ARA_MW_ is a potential biomarker that may stratify BAT responders from non-responders [88,95,96,97].

BAT is well tolerated. Most adverse events (AEs) are low grade and well tolerated (fatigue, musculoskeletal pain, edema, nausea, breast tenderness). Grade 3 to 4 AEs rarely described are hypertension, pulmonary embolism, and back pain [15].

### 5.2. Other Agents Capable of Resensitizing CRPC to Antiandrogen Therapy

#### 5.2.1. Niclosamide

Targeting AR variant expression is one way to achieve restoring sensitivity to antiandrogens [98].

Niclosamide, an anti-helminthic drug, has demonstrated preclinical efficacy in reducing AR-V7 expression in enzalutamide-resistant cells [99,100].

Liu et al. established that niclosamide could induce AR-V7 protein degradation and reduce AR-V7 recruitment to promoter regions of target genes, resulting in reduced transcriptional activity and thus resensitizing resistant cells to enzalutamide and abiraterone treatment.

The combination of niclosamide and either enzalutamide or abiraterone produced maximal tumor inhibition in a xenograft model.

After that, they demonstrated that a combination of niclosamide and bicalutamide overcomes treatment resistance to enzalutamide in vitro [101].

Based on these preclinical data, a phase I/II study established the safety of abiraterone in combination with niclosamide in CRPC patients [102].

However, these findings differ from the study phase I, which evaluated the combination of enzalutamide and niclosamide in patients with mCRPC finding high toxicity [103].

Another mechanism by which niclosamide could reverse resistance to enzalutamide is through the downregulation of the androgen receptor-STAT3 signaling axis [104].

#### 5.2.2. Monoamine Oxidase A Inhibitors (MAOAIs)

MAOAIs have been shown to inhibit the growth and development of PCa metastasis due to high concentrations of MAOA in the prostate. MAOAI combined with enzalutamide synergizes and sensitizes cells to enzalutamide treatment [105].

#### 5.2.3. Notch Receptors

Inhibitors of gamma-secretase, which regulates cleavage of cell surface receptors, including Notch receptors, have been demonstrated to inhibit PCa cell growth in vitro in combination with enzalutamide or abiraterone and are also capable of resensitizing enzalutamide-resistant cells and xenografts to enzalutamide treatment [106].

#### 5.2.4. Indomethacin

Indomethacin, a nonsteroidal anti-inflammatory drug, acts as an inhibitor of AKR1C3 enzymatic activity and, when elevated, is linked to enhanced PCa progression, aggressiveness, and resistance to antiandrogens and radiation therapy [107], and it has been established that that may restore enzalutamide and abiraterone sensitivity in resistant PCa cells by targeting intracrine androgens [108,109].

Based on these preclinical studies, a single-arm phase II trial showed the efficacy and tolerability of the indomethacin and enzalutamide combination to treat CRPC [110].

#### 5.2.5. Lapatinib

Abiraterone-resistant xenograft models have increased ErbB2 activity. It has been demonstrated that lapatinib blocks ErbB2 and, in combination with abiraterone, can enhance treatment response [111].

Furthermore, it has also been established that lapatinib can enhance enzalutamide response by inhibiting the HER2 signaling axis [112].

#### 5.2.6. Panobinostat

Histone deacetylases (HDACs) are required for AR mRNA transcription and protein stability and transcriptional activation of AR target genes. HDAC inhibitors (HDACIs) are promising anticancer drugs targeting epigenetic modes [113].

Panobinostat, an HDACI, has been proven to reduce AR mRNA levels, including ARV7, as well as cell proliferation in vitro and tumor growth in vivo in all CRPC model systems [114].

Unfortunately, these promising preclinical observations have yet to translate into beneficial clinical activity of HDACI monotherapy in CRPC, maybe because low concentrations were used due to its toxicity.

However, researchers found an alternative use for panobinostat when they discovered its ability to overcome the prevalent acquired resistance to antiandrogenic agents in CRPC.

They demonstrated a synergetic inhibition of PCa cell growth between panobinostat and bicalutamide in culture cells resistant to bicalutamide [115], and what is more interesting, they found that panobinostat can resensitize CRPC to the antiandrogen they became resistant to, rewriting the epigenetic code [116]. They also noted that frequent and intermittent exposure to panobinostat in combination with the antiandrogen was required due to the reversibility of HDACI-induced epigenetic changes [117].

These promising results of panobinostat in combination with bicalutamide encourage authors to design new trials in combination with more powerful antiandrogen enzalutamide, which also works by binding to the AR LBD and develops resistance linked to AR overexpression and ARV7.

#### 5.2.7. EPI-7386

As explained before, AR-V7 expression after enzalutamide or abiraterone therapy correlates with a high percentage of resistance to treatment and worse response [38]. Therefore, disruption of AR-V activity might be a way to reverse resistance to these treatments.

EPI-001 is a bisphenol A-derived antagonist that binds the AR N-terminal domain to block the transcriptional activity of AR and several AR-Vs, including AR-V7. It has been shown that EPI can inhibit the proliferation of enzalutamide-resistant cells in vivo and in vitro [118]. Based on these studies, a phase I/II clinical trial (NCT02606123) was designed, with a prodrug formulation of the compound, EPI-506, in CRPC patients progressing to abiraterone and/or enzalutamide.

Unfortunately, it showed minor PSA declines, bringing to light the need for more potent and metabolically stable NTD inhibitors.

The next generation of NTD inhibitors, EPI-7386, is more active and metabolically stable than EPI-506 [119,120], and its ability to revert resistance to enzalutamide is being studied in a phase1/2 clinical trial that compares EPI-7386 in combination with enzalutamide vs. enzalutamide alone (NCT05075577) [121].

#### 5.2.8. Autophagy Inhibitors

Autophagy is a process associated with drug resistance in different cancers. It is an adaptative response to maintain cell survival under metabolic stress, including androgen deprivation. It is an essential mechanism of resistance to ARSis in CRPC [46].

Although the exact mechanism remains unknown, ADT has been shown to promote autophagy.

Clomipramine and metformin are autophagy inhibitors and have proven to overcome enzalutamide resistance in vitro and in vivo, significantly reducing tumor growth in enza-resistant prostate cancer cells [122]. In addition, metformin improves the efficacy of both abiraterone and enzalutamide in vitro [123].

#### 5.2.9. Antisense Oligonucleotides (ASO)

Epigenetic modifications in CRPC are being studied, specifically epigenetic inhibitors, for their ability to render CRPC cells sensitive to enzalutamide.

Numerous investigations are focused on targeting epigenetic mediators since it is known that epigenetic aberrations are associated with PCa progression and drug resistance [124].

Antisense oligonucleotides (ASO) inhibit the histone lysine-N-methyltransferase (EZH2), which correlates with poor prognosis in PCa.

Xiao et al. observed that ASO may have a strong influence on the AR pathway, causing redistribution of AR binding and upregulated AR signaling, and demonstrated that treatments that inhibit EZH2 may resensitize tumor cells to AR-targeting treatments in CRPC [125]. In fact, it had already been shown that this type of drug could be effective in NEPC by reactivating AR signaling and, therefore, its sensitivity to antiandrogens [126,127].

Based on all these findings, ASO is postulated as a possible strategy to restore sensitivity to antiandrogen therapy.

#### 5.2.10. Bet Pathway

The bromodomain and extra-terminal (BET) family proteins can regulate DNA transcription. Some proteins of this family can interact with AR in the nucleus, acting as a mediator of AR and AR-Vs transcription.

The inhibition of BET pathways is being studied as an attractive option to overcome resistance to androgen inhibition [62,63]. The compounds being studied include the BET inhibitor ZEN003694 and the BRD4-targeting compound JQ1 [128,129].

#### 5.2.11. Dasatinib

The SRC gene encodes the proto-oncogene tyrosine protein kinase Src, and there is increasing evidence about its correlation with AR pathways and the development of castration-resistant status. Src can favor the development of CRPC by sensitizing AR to intracrine androgen levels by developing canonical and non-canonical AR binding site-associated genes. Src inhibitor dasatinib has been demonstrated to reduce the expression of the splice variant AR-V7, helping to overcome antiandrogen resistance [130,131].

#### 5.2.12. Ipatasertib

Ipatasertib is an Akt inhibitor that has been demonstrated to resensitize CRPC cells to antiandrogens when combined with enzalutamide, inducing apoptosis and leading to remarkable tumor cell growth inhibition, both in vitro and in vivo [132,133].

Table 1 summarizes the agents and strategies that can resensitize CRPC cells to antiandrogen therapy and their mechanisms of action.

## 6. Conclusions

Even in the advanced stages of the disease, AR is the key therapeutic target of PCa.

For this reason, several studies focus on an alternative concept beyond the inhibition of tumor growth.

This concept focuses on the ability of different agents to resensitize CRPC cells to previously used antiandrogen treatments that have lost their effectiveness, allowing them to overcome antiandrogen resistance and achieving prolonged responses in multidrug-treated patients.

It has been demonstrated that BAT can restore sensitivity to ARS is in CRPC patients. These findings represent a major step toward this use of BAT, although since not all patients respond uniformly to this maneuver, it is essential to determine predictors of response and resistance mechanisms to this interesting therapy.

In this vein, other agents have been studied, and studies are at different stages of development. In any case, these findings open the door to the interesting possibility of using specific drugs as a hinge to effectively reuse treatments for which the tumor had already developed resistance.

## Figures and Tables

**Figure 1 biomedicines-11-01105-f001:**
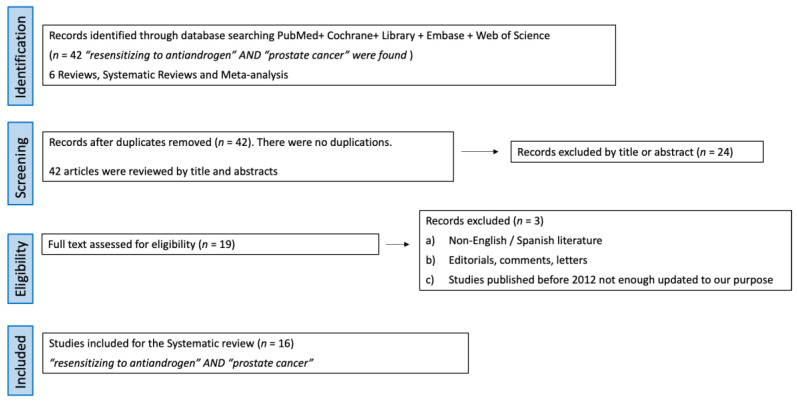
Preferred reporting items for Systematic Reviews and Meta-analyses (PriSMa) criteria and population, intervention, comparator, outcomes (PICO) methodology applied to the article.

**Figure 2 biomedicines-11-01105-f002:**
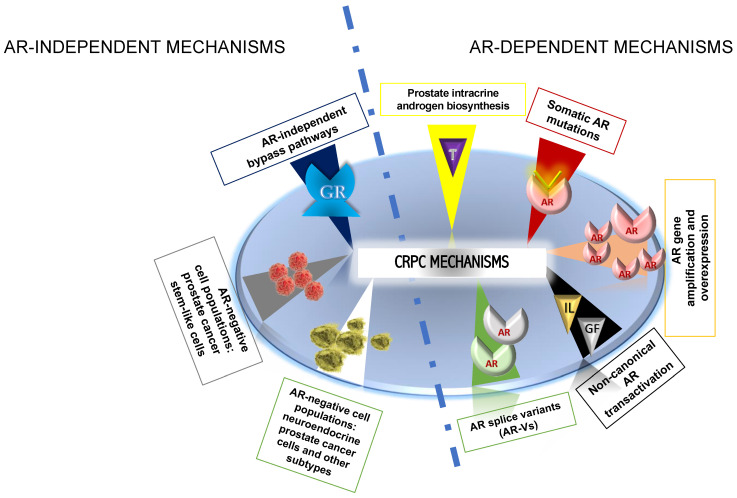
Mechanisms of resistance to androgen deprivation therapy. T, testosterone; AR, androgen receptor; IL, interleukin; GF, growth factor; GR, glucocorticoid receptor.

**Figure 3 biomedicines-11-01105-f003:**
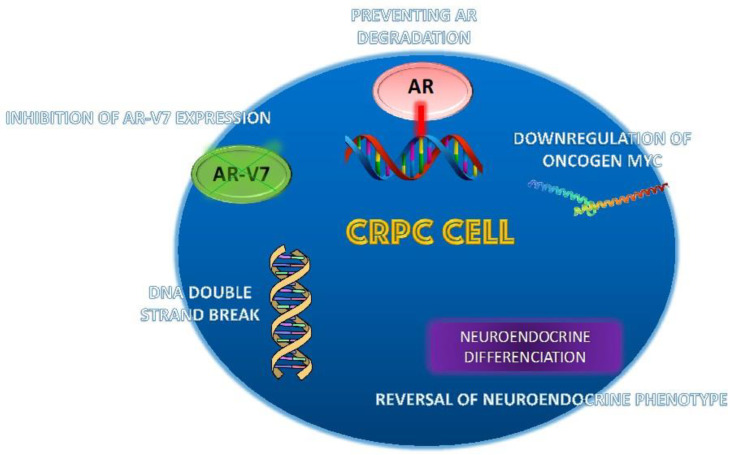
Proposed mechanisms of action of bipolar androgen therapy (BAT). Different mechanisms have been proposed—Induction of genomic instability through DNA double-strand breaks—Stabilization of the link between DNA and the AR, preventing AR degradation and DNA relicensing during the cell cycle—Modulation of oncogene expression (downregulation of oncogene Myc)—Inhibition of AR-V7 expression—Reversal of transition to the neuroendocrine phenotype.

**Figure 4 biomedicines-11-01105-f004:**
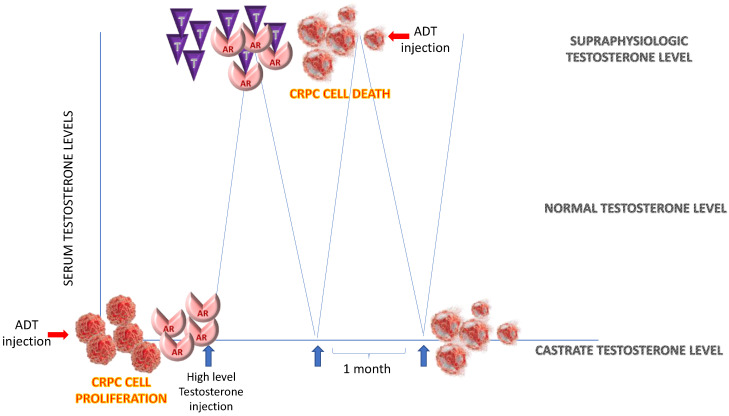
Antitumor mechanism of intermittent administration of high-dose androgens in CRPC cells. Development of castrate-resistant tumor cells involves acquisition of adaptive auto-regulation giving place to an increase in AR expression in a low androgen environment. This increase inhibits apoptotic death and stimulate cancer cell proliferation. AR is needed for DNA replication in CRPC cells. When these cells are acutely exposed to supraphysiologic androgen, AR is saturated. This induces a paradoxical cancer cell death via over-stabilizing AR, preventing full relicensing in the next cell cycle.

**Figure 5 biomedicines-11-01105-f005:**
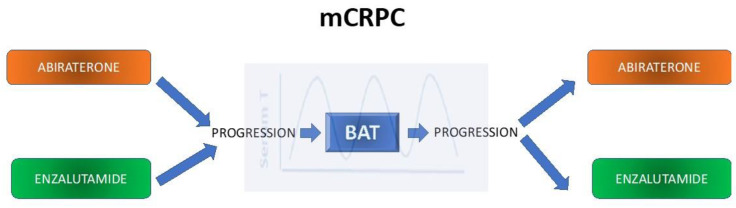
Scheme of the main studies showing BAT’s ability to resensitize to subsequent ARSi in mCRPC.

**Table 1 biomedicines-11-01105-t001:** Agents and strategies that can resensitize CRPC cells to antiandrogen therapy and their mechanism of action.

*Drug*	Target	Mechanism of action	Reference
***BAT*** *(high level of testosterone followed to castrate range)*	Androgen Receptor (AR)	- DNA double strand breaks - Prevents AR degradation - Inhibition of AR-V7 expression - Repression of SKP2 and MYC	Schweizer, MT. Sci Transl Med, 2015 [77] Benjamin AT. Lancet Oncol, 2018 [78] Markowski, MC. Eur Urol, 2021 [79] Denmeade, SR. J Clin Oncol, 2021 [81]
** *Niclosamide* **	Androgen receptor (AR)	- Reduce expression of AR-V7 - Down regulation of the androgen receptor-STAT3 signaling axis	Liu, C. Mol Cancer Ther, 2017 [101] Liu, C. Prostate 2015 [104]
** *Monoamine oxidase A inhibitors (MAOAIs)* **	Monoamine oxidase A in prostate	Sensitizes cells to enzalutamide	Gaur, S. Prostate, 2019 [105]
**Inhibitors of Notch receptors**	Notch receptors	Inhibition of gamma secretase	Rice, MA. Mol Cancer Ther, 2019 [106]
** *Indomethacin* **	Intracrine androgens	Inhibition of AKR1C3 enzymatic activity	Liu, C. Cancer Res, 2015 [108]
** *Lapatinib* **	HER2 signaling axis	- Blocks ErbB2 activity - Inhibition of HER2	Shiota, M. Oncotarget, 2015 [112]
** *Panobinostat* **	Histone deacetylases (HDAC) pathway	HDAC inhibition	Ferrari, AC. Clin Cancer Res, 2019 [115]
** *EPI-7386* **	AR N-terminal domain	Inhibition of the N-terminal domain (NTD) of the AR	Hong, NH. Cancer Res, 2020 [120]
** *Clomipramine and metformine* **	Autophagy inhibition	Autophagy inhibition	Nguyen, HG. Oncogene, 2014 [122]
** *Antisense oligonucleotides (ASO)* **	Epigenetic modifications	Inhibition of the histone lysine-N-methyltransferase EZH2	Xiao, L. Cancer Res, 2018 [125]
** *ZEN003694/JQ1* **	Bromodomain and extra-terminal (BET) pathway	Inhibition of BET and BRD4	Asangani, I.A. Nature, 2014 [129]
** *Dasatinib* **	SRC gene	inhibition of proto-oncogene tyrosine protein kinase Src	Chattopadhyay, I. Oncotarget. 2017 [131]
** *Ipatasertib* **	PI3K-Akt-mTOR pathway	Akt inhibitor	Adelaiye-Ogala, R. Mol Cancer Ther, 2020 [133]

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
