# Peer review of "Strategies to Re-Sensitize Castration-Resistant Prostate Cancer to Antiandrogen Therapy"

_biomedicines, 2023, doi:10.3390/biomedicines11041105_

Round 1

Reviewer 1 Report

Overall well written and comprehensive overview of this important topic. Just needs a careful proofreading as there are several typos and minor grammatical errors.

Author Response

Dear reviewer,

Thank you very much for your contributions. As you suggest, the text has been carefully revised to correct gramatical errors. I send you a copy of the new text with all the changes added.

Thank you so much

Reviewer 2 Report

                In this review, Ruiz et al. summarize recent strategies to re-sensitize castration resistant prostate cancer (CRPC) to anti-androgen therapies. This is a well-written, objective and comprehensive review that both summarizes and evaluates  newer approaches to therapy of CRPC.   The language is clear and the organization of the manuscript is logical. The Tables and Figures are excellent and enrich the text. There is special emphasis on the use of  Bipolar Androgen Therapy  (BAT) in CRPC treatment and an interesting discussion of  possible mechanisms of action including the induction of double strand DNA breaks. The review should be of interest both to experienced investigators and those new to the field.

A few minor suggestions are as follows:

A guide to abbreviations would be helpful.

Lines 21-23 could be incorporated into the text rather than shown as bullet points.

Line 71-72 Please provide a reference.

In the discussion of autophagy and metformin, the authors should reference the paper by Xie et. al. Cancers 13:633, 2021

The manuscript should be copy-edited for some typographical errors (i.e. line 542)

Author Response

Dear reviewer,

Thank you very much for your contributions. As you suggest, the text has been carefully revised and all your suggestions have been incorporated. I send you a copy of the new text with all the changes added.

Thank you so much

Reviewer 3 Report

The development of a castration-resistant status represents a main issue during androgen deprivation therapy for prostate cancer.

In the literature, various studies have investigated the use of  new drugs that act with a different mechanism of action, combination therapies or strategies that resensitize tumors to antiandrogens.

In the present paper, the authors sought to provide a review on the mechanisms that allow cancer cells to switch to an androgen receptor-negative disease and potential strategies to successfully re-sensitize these CPRC cells to antiandrogen therapy.

Title: accurate

Abstract: reflects the report

Introduction:  clearly states background.

Results:  a Table summarizing data from main included studies would be very useful for readers

Conclusions: they should be more concise

Author Response

Dear reviewer,

Thank you very much for your contribution. Your suggestion is now in the text.

Best regards

Reviewer 4 Report

Title: Strategies to re-sensitize castration resistant prostate cancer to
antiandrogen therapy
Authors: Belén Congregado Ruiz, Inés Rivero Belenchón, Guillermo Lendínez Cano, Rafael Antonio Medina López Summary:

In this article, the authors have attempted to identify the mechanisms that allow cancer cells to transition from androgen receptor-positive to androgen receptor-negative disease, and they focus on possible strategies to overcome these mechanisms and successfully re-sensitize these castration-resistant prostate cancer cells to anti-androgen therapy. Of all the mechanisms of re-sensitization to anti-androgens, bipolar androgen therapy has been shown to be the most effective. Therefore, they reviewed the most recent literature on bipolar androgen therapy and data on other new agents.

Several major points are listed below:

1: The abstract is too short, not properly formatted and should be rewritten as it appears unstructured. The most important alternatives should be written numerically one after the other in the text.

2: The novelty of the article should be emphasized more clearly throughout, which is not the case here.

3: The introduction should be restructured from scratch. Much more should be written about antiandrogenic therapy, prostate cancer, and BAT.

4: The search strategy used for the literature search should be stated.

5: The limitations associated with the use of BAT should be discussed.

6: I would suggest that the authors create a nice table on AR-dependent and independent mechanisms. This would add more weight to the paper.

7: I miss a general summary discussion.

8: All abbreviations should be defined at their first mention and used thereafter.

9: Please use consistent abbreviations throughout the manuscript.

10: The paper contains many complicated abbreviations. A list of the most important abbreviations would be helpful to the reader.

11: The English language in general needs improvement. The entire text needs to be revised by a native English speaker.

Author Response

Dear reviewer,

Thank you very much for your contribution.

All your suggestion have been now incorporated into the text.

Best regards

Round 2

Reviewer 4 Report

The authors have addressed all points of potential criticism and each suggestion made by the reviewer adequately and in detail.